# Sustainable Electroporator for Continuous Pasteurisation: Design and Performance Evaluation with Orange Juice

Rai Naveed Arshad [1], Zulkurnain Abdul-Malek [1,*], Yanti M. M. Jusoh [2], Emanuele Radicetti [3], Paola Tedeschi [3], Roberto Mancinelli [4], Jose M. Lorenzo [5,6] and Rana Muhammad Aadil [7,*]

1 Institute of High Voltage and High Current (IVAT), School of Electrical Engineering, Faculty of Engineering, Universiti Teknologi Malaysia, Johor Bahru 81310, Malaysia; rainaveed@yahoo.co.uk

2 Food and Biomaterial Engineering Research Group, School of Chemical Engineering, Faculty of Engineering, Universiti Teknologi Malaysia, Johor Bahru 81310, Malaysia; yantimaslina@utm.my

3 Department of Chemical, Pharmaceutical and Agricultural Sciences (DOCPAS), University of Ferrara, 44121 Ferrara, Italy; emanuele.radicetti@unife.it (E.R.); paola.tedeschi@unife.it (P.T.)

4 Department of Agricultural and Forestry Sciences (DAFNE), University of Tuscia, Via S. Camillo de Lellis snc, 01100 Viterbo, Italy; mancinel@unitus.it

5 Centro Tecnológico de la Carne de Galicia, Rúa Galicia No 4, Parque Tecnológico de Galicia, 32900 Ourense, Spain; jmlorenzo@ceteca.net

6 Área de Tecnología de los Alimentos, Facultad de Ciencias de Ourense, Universidad de Vigo, 32004 Ourense, Spain

7 National Institute of Food Science and Technology, University of Agriculture, Faisalabad 38000, Pakistan

* Correspondence: zulkurnain@utm.my (Z.A.-M.); muhammad.aadil@uaf.edu.pk (R.M.A.)

**Abstract:** Electroporation is a simple but effective and sustainable food processing way of treating cell membranes with an electric field. It is employed in a variety of ways in the food industry, ranging from shelf-life extension to green extraction. Despite its wide range of applications, electroporators are out of reach for many labs due to their high development costs, and different electroporators have been tailored to specific applications. The designing sequence of an electroporator that takes the geometry of a treatment chamber and its electrical resistance into account for the design of a pulse generator has not been addressed in published literature. To meet this demand, this study presents a straightforward way to develop a simple, affordable, and portable electroporator for liquid food pasteurisation. The proposed electroporator comprises a coaxial treatment chamber with static mixers and a high-voltage Marx bank based on insulated-gate bipolar transistors (IGBTs). The generator has a 4.5 kV output voltage and a peak current rating of 1 kA; however, the modular design allows for a wide range of voltage and current ratings. Treated orange juice using thermal pasteurisation (65 °C, 30 min) was also used for comparison. The performance of the electroporator was studied using chemical and microbial tests. A significant log reduction (5.4 CFU·mL$^{-1}$) was observed in both the PEF-treated samples with sieves. Additionally, the treated juice visual and chemical color analysis showed that the PEF-treated sample extended the shelf-life after 9 days of storage at 4 °C. This research also examines the energy conversion in these two processing steps. This study assists in developing further electroporators for other food applications with different treatment chambers without compromising the product's quality.

**Keywords:** pulsed electric field; sustainable technology; nonthermal pasteurisation; solid-state marx generator; energy efficient; food safety

## 1. Introduction

The food industry's future economic growth and prosperity depend on the adoption of creative, sustainable, and innovative processing technologies to produce high-quality foods with extended shelf-life. Pulsed electric field (PEF) is incredibly energy efficient and plays a significant part in this technological revolution. An external electrical field is utilized in PEF treatment; ions' inner and outer living cells align along the field. The

attraction of opposing charges on both sides of the membrane results in compression of the membrane thickness [1,2]. Thus, an intensive electric field changes a trans-membrane potential that leads to the breakage of the lipid membrane [3]. This process, electroporation, alters the membrane's phospholipid bilayer, resulting in the loss of viability and cell death. The juice industry is one of the sectors where it is being used [1].

The development of the PEF system and creating an optimum treatment protocol to ensure the safety of foods have been interesting topics of study over the past decade [4]. Figure 1 shows a typical electroporator for PEF processing of liquid food that comprises one or more treatment chambers, as well as a high-voltage pulse generator [5]. A pulse generator must have high reliability for continuous operation and a low cost in PEF treatment [6]. Hence, some researchers have designed different pulsed generators for PEF treatment [7–9]. Each design goal effectively implemented the PEF system at the lowest possible cost while maintaining operator safety [10]. High voltage pulse generator setups are often based on electrical circuits that utilize a capacitor discharge through a high-power switch. This capacitor is discharged using a mercury-wetted reed relay, a spark-gap switch, or a semiconductor power switch [5]. The majority of existing PEF setups are created by high-tech laboratories that need operators with substantial training and specialized knowledge. They also cost a lot, which has slowed down the use of this technology in food research. As a result, many compact electroporators needed to be developed for specific laboratory needs.

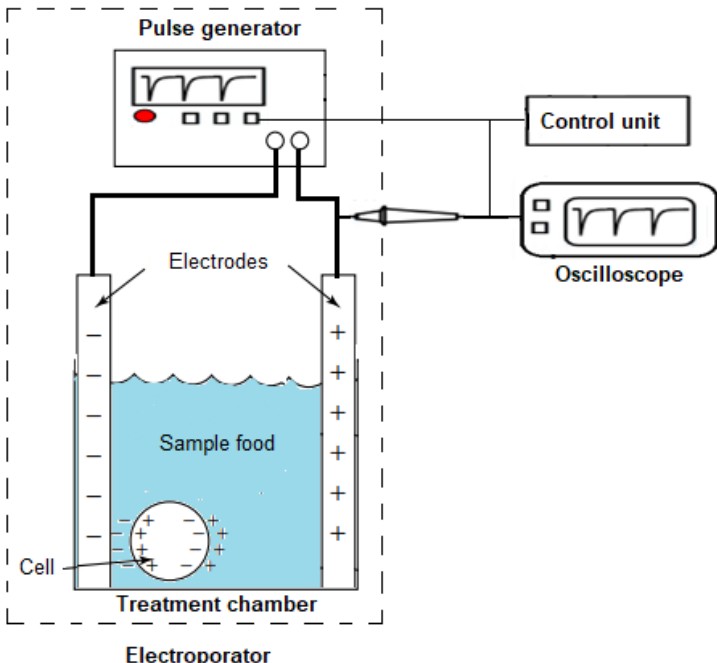

**Figure 1.** A schematic diagram of a general electroporator for liquid food treatment.

The success of PEF-based food processing is dependent on the treatment chamber. The treatment chamber design influences peak field intensity, treatment uniformity, and process efficiency. The treatment chamber must be able to match the pulser's impedance. Several designs of static and continuous treatment chambers have been suggested. Coaxial treatment chambers effectively provide precisely defined electric field distributions for medium-sized volumes [11]. However, some measures must be taken to maintain resistance and minimum electric field variations in its treatment zone [12]. Static mixing is a technique extensively used in the process industry, and its significance in PEF pasteurisation has gone unnoticed. It helps overcome laminar liquid flow in the treatment zone and control temperature rise [13]. Only a few research papers have simulated this idea with a collinear treatment chamber [13–16]. It has, however, never been utilized with a coaxial arrangement, nor has it ever been employed and tested for treating liquid fluids, since the load may vary

from a few ohms to hundreds of ohms depending on the treatment chamber's geometry and electrical conductivity of the food sample. This implies that the pulse generating circuit must be built such that the produced pulse waveform is insensitive to the load value.

This study aims to present the design methodology for a simple-to-use, compact electroporator for liquid food pasteurisation that can be modified for various loads (different conductivity of the sample food). The proposed electroporator is composed of a continuous coaxial treatment chamber, and an insulated gate bipolar transistor (IGBT)-based Marx generator. Furthermore, two insulator sieves were inserted into the treatment zone to disturb the laminar flow. Additionally, it presents a complete procedure to develop the high-voltage generator required for the designed treatment chamber.

## 2. Materials and Methods

Before developing the necessary high-voltage pulse generator, the chamber's design must be thoroughly investigated and evaluated for electrical resistance and the electrical field distribution inside the treatment zone should be determined [16,17].

### 2.1. Analysis and Fabrication of a Coaxial Treatment Chamber Prototype

In a coaxial chamber, the electric field is affected by the radial location inside the chamber. The electrical field intensity is greatest at the electrode area in the coaxial treatment chamber connected to the high voltage source [18]. The following equations determine the field strength inside at '$r$' ($R_i < r < R_o$) and electrical resistance:

$$E = \frac{V}{r \ln\left(\dfrac{R_o}{R_i}\right)} \tag{1}$$

$$R = \frac{\ln\left(\dfrac{R_o}{R_i}\right)}{2\pi\sigma l} \tag{2}$$

where $R_i$ and $R_o$ are the inner and outer electrodes' radii, respectively, '$l$' is the length of the treatment zone, and the sample food conductivity '$\sigma$' consists of decisive parameters in designing the treatment chamber's dimensions.

It indicates that lowering the spacing between electrodes reduces electric field heterogeneity. However, it has a negative impact on the volume of a huge industrial facility. Thus, to keep the chamber's volume modest, the proposed design uses an outer electrode with a radius of 3.0 cm. Similarly, a long treatment chamber is usually beneficial for extending treatment duration and preventing extended periods of unwanted overheating [19]. However, as the length of the treatment zone grows, the resistance of the treatment chamber decreases. As a result, a higher-current power supply is needed, which adds to the supply design issues [5]. The chamber's resistance rises when the electrode spacing and radii are increased, and it conflicts with the radial variations in the strength of the electric field. Consequently, a design must be selected that satisfies these two constraints. These measurements have been determined to meet these requirements: $l$ = 3 cm, $R_i$ = 2.5 cm and $R_o$ = 3 cm. A resistance of 5 ohms has been calculated for the treatment chamber, presented here with orange juice (conductivity: 0.289 S/m).

Two circular insulator rings (radii of 3.0 and 3.5 cm) consisting of tiny holes (0.15 mm) were used as a static mixer (sieves). According to the sample food, these sieves were detachable, permitting replacement for variation of the mesh grids. In addition, these insulated sieves were placed perpendicular to the flow direction of the treated liquid sample. Figure 2 depicts the mechanical assembly of the treatment chamber.

Stainless steel electrodes were used, whereas Teflon was used for the insulating base (Polytetrafluoroethylene—PTFE). Since acrylic is transparent and lightweight, it was chosen for both the mesh and the outside casing of the coaxial chamber.

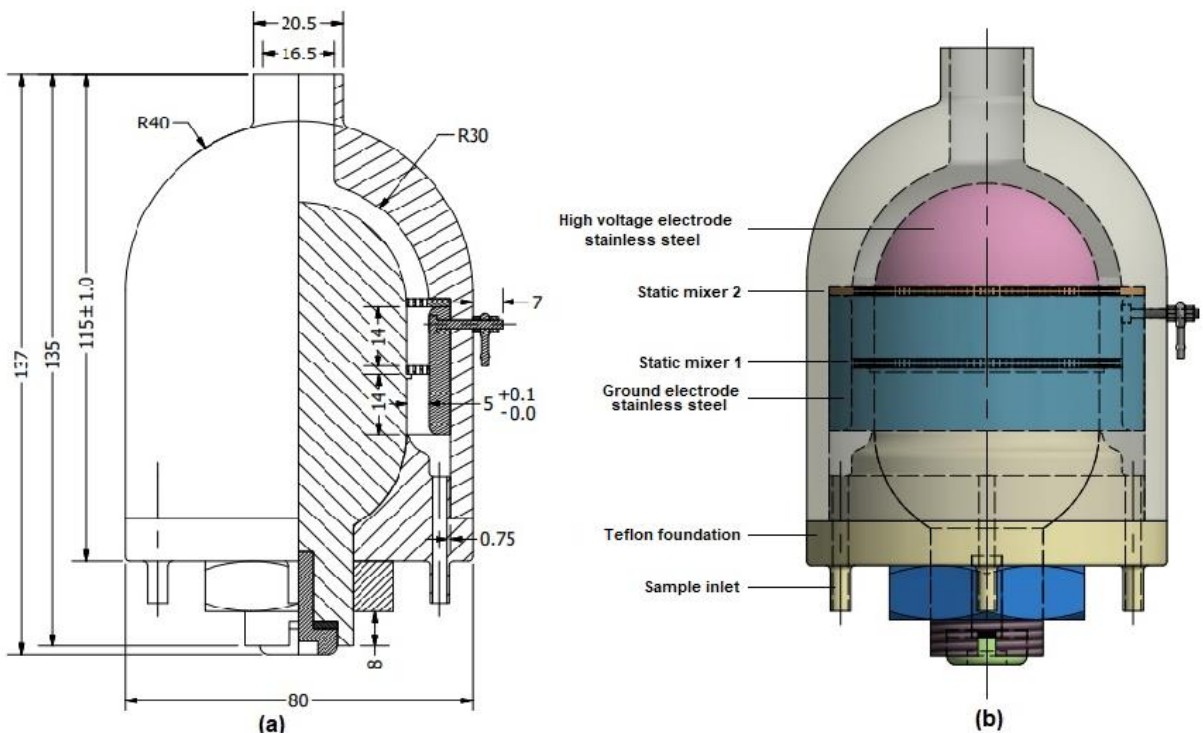

**Figure 2.** Two-dimensional mechanical model of the proposed coaxial treatment chamber. (**a**) Dimensions in mm, (**b**) different parts.

### 2.2. Designing a Marx Generator for Food Electroporation

After determining the size of the treatment area, the following section details the generator's design parameters. Because all parameters are interconnected, and changing one may impact the others, it is necessary to compute each parameter separately. Based on the literature review, the parameters listed in Table 1 are needed for PEF pasteurisation and are also required during the system's design [20].

**Table 1.** Parameters for designing a pulse generator.

| Parameter | Value | Units |
|---|---|---|
| Required electric field | 10 | $kV \cdot cm^{-1}$ |
| Pulse width | 3 | $\mu sec$ |
| Maximum flow rate | 100 | $mL \cdot sec^{-1}$ |
| Required energy per volume (specific energy) | 200 | $J \cdot mL^{-1}$ |
| Chamber volume | 25.8 | $cm^3$ |
| $3.14 \ (3^2 - 2.5^2) * 3$ | 25.8 | mL |

Equation (1) gives the required output voltage for the PEF pasteurisation with the proposed coaxial geometry:

$$V_{out} = 10 \ kV \cdot cm^{-1} * 2.75cm * \ln(3/2.5) \approx 4.5 \ kV \tag{3}$$

For the designed coaxial treatment chamber containing orange juice, the equivalent resistance was calculated as 5 $\Omega$.

$$I_{out} = V_{out}/R \approx 1 \ kA \tag{4}$$

As 1 joule = 1 ampere * 1 volt * 1 s, the energy delivered per pulse can be found with the following equation:

$$\text{Energy per pulse} = V_{out} * I_{out} * PW = 4.5 \text{ kV} * 1 \text{ kA} * 3 \text{ μs} = 13.5 \text{ J/pulse} \tag{5}$$

The system's total energy ($J \cdot sec^{-1}$) can be determined using the energy applied per unit of volume and the liquid's flow rate.

Total energy·$sec^{-1}$ = Required energy.$volume^{-1}$ * Max. flow rate = 20 kJ·$sec^{-1}$

The total energy per unit of volume and energy per pulse can be used to find the required frequency. Then, the repetition rate (Hz) and the number of pulses are calculated using the energy value and the size of the chamber.

Frequency = Total energy·$second^{-1}$/Energy·$pulse^{-1}$

Frequency = 20 kJ·$sec^{-1}$/15 J $pulse^{-1}$ = 1.3 kHz

The residence time of the sample food inside the treatment zone can be computed as follows:

Residence time (sec) = Liquid volume in treatment zone/liquid flow rate = 26 mL/ 100 mL ·$sec^{-1}$ = 0.258 sec

Number of pulses = Residence time (sec) * frequency (Hz) = 0.258 sec * 1.3 kHz = 335

Energy storage in a capacitor is delivered as an energy per pulse, and it is used to calculate the total capacitance of the Marx generator as follows:

Energy per pulse = 0.5 * C * $V_{out}^2$

15 = 0.5 * C $(5 \text{ k})^2 \approx 1.5$ μF

Some safety margin has been introduced into the total capacitance.

Each stage's capacitance should be $3 \times 1.5$ μF $\approx 4.5$ μF, which results in a total capacitance of 1.5 μF. It was assumed that these capacitors would have to endure a minimum input voltage of 2 kV, including the safety margin. The capacitors were chosen with a series leaking inductance of 19 nH. The discrete IGBTs BiMOSFET IXBX55N300 made by the IXYS Company (Leiden, Netherland), with ratings of 3 kV and 120 A, was chosen for this application. Turning on IXBX55N300 requires the gate charge ($Q_G$ = 335 nC; datasheet IXBX55N300), and assuming a rise time of 50 ns for the switch, the maximum peak current needed at the output of the gate drive was calculated using the following formula:

$I_G = Q_G$/transition = 335 nC/50 nsec = 6.7 A

The Microchip company manufactures the TC4422, which was selected to meet the requirement of the driver chip for the selected IGBT. The maximum output current is 9 A, and the supply voltage ranges from 4.5 V to 18 V. There is latch-up protection for a chip that can handle up to 1.5 A of reverse current. Figure 3 shows a block diagram of a switching board of the proposed Marx generator.

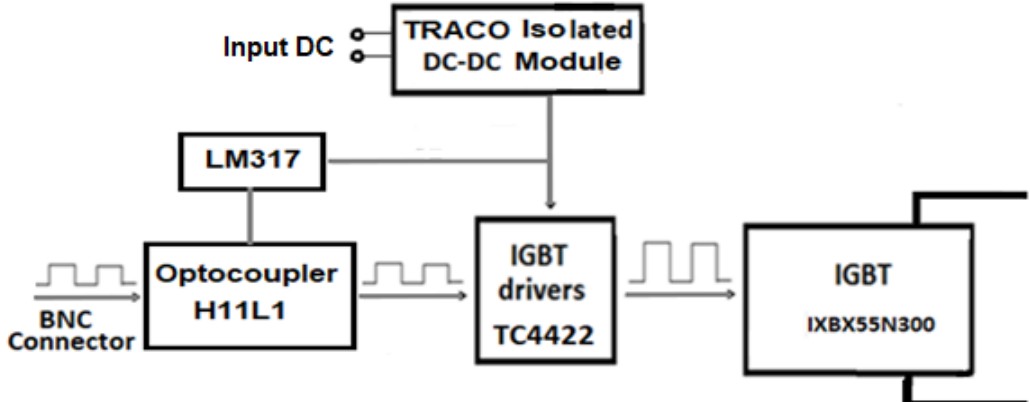

**Figure 3.** Block diagram of a switching board in the proposed Marx generator.

Each stage of the Marx generator was composed of a single switching board, and these switching boards were linked with other boards through wiring. Nylon PCB spacers were

used to lift one board over another, which enabled the switching boards to be replaced or removed quickly. Each board was composed of two parallel IGBT (IXBX55N300), two driver chips (TC4422), an optocoupler IC (H11L1), an isolated dc-dc converter (TEN3-4813N), and an LM317. The optocoupler provided the isolation by using light and delivered a TTL signal (command signal) to the driver IC. Driver IC switched IGBT on and off, following the command signal. The board's power was delivered by isolated dc-dc converters, which separated the whole board from the ground potential. On a printed circuit board (PCB), these components and other supporting circuit components were soldered. An IGBT IXBX55N300 can provide a peak current of 800 A during a pulsed operation. Therefore, two parallel IGBTs at each stage were coupled to increase the current delivered to the load. It was also found that paralleling two IGBTs in each stage reduces the loss factor by reducing the current flow through each device. Figure 4 shows a developed three-stage Marx generator.

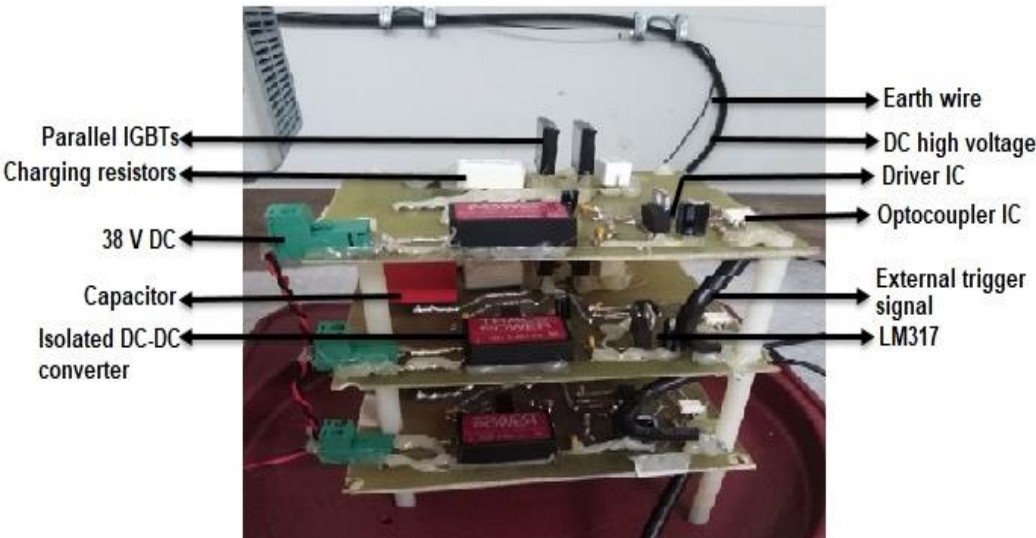

**Figure 4.** A three-stage Marx generator designed for electroporation.

The gate current of IGBT and MOSFET devices, which are designed to be controlled by voltage pulses, is used in reality to regulate the switching speed of these devices. The input capacitor of these devices must be charged fast in order to increase the switching speed. The driver chip TC4422 (9 A) performance was tested for IGBT switches IXBX55N300 (3 kV, 55 A). For this test, a resistance of 5 $\Omega$ was selected as a load. Figure 5 shows the voltage at the gate of the IXBX55N300 chip that the TC4422 chip drives. It delivers $260 \pm 40$ ns from 10–80% of 18 V. Another test was performed utilizing two gate driver chips to drive a single IGBT at a higher switching speed. However, increasing current capability has no discernible impact and does not appreciably enhance the rising time. Since the IGBT gate has required even less current than 9 A, which a single gate driver IC can easily provide, two driver chips could increase the rise time by 40 ns. This improvement was not significant to use the multiple-gate triggered IC.

Figure 6 shows the oscilloscope snapshot of the output waveform. It produced a waveform with an amplitude of $4.2 \pm 0.2$ kV and a width of 4 μs across a load of 5 $\Omega$. Prototype circuitry significantly impacts parasitic inductance and capacitance, as the output pulse waveform has voltages that go above and below the nominal voltage.

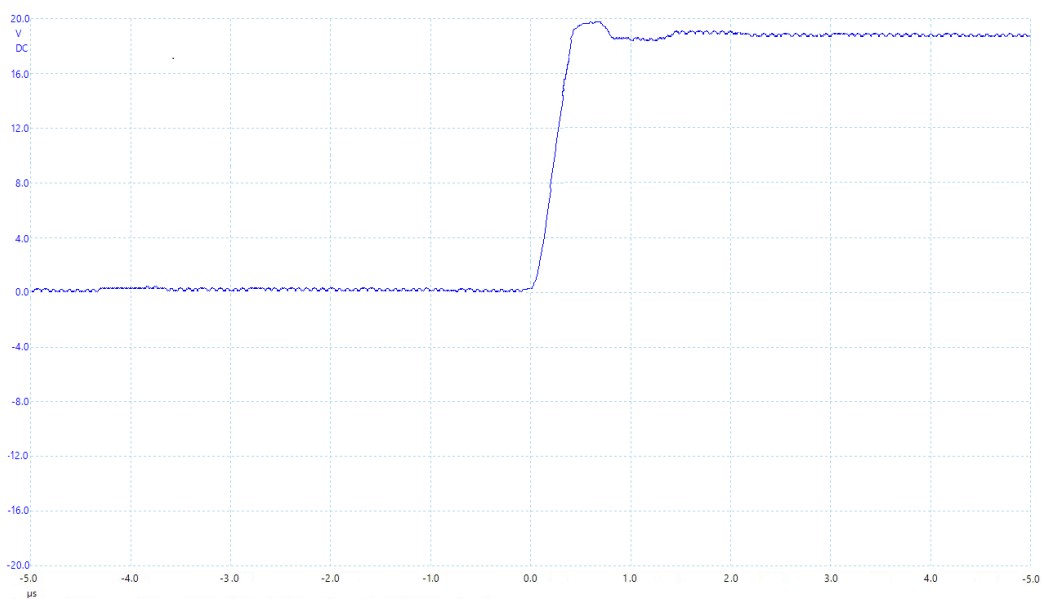

**Figure 5.** The rising edge of a trigger signal from two TC4422 gate driver ICs.

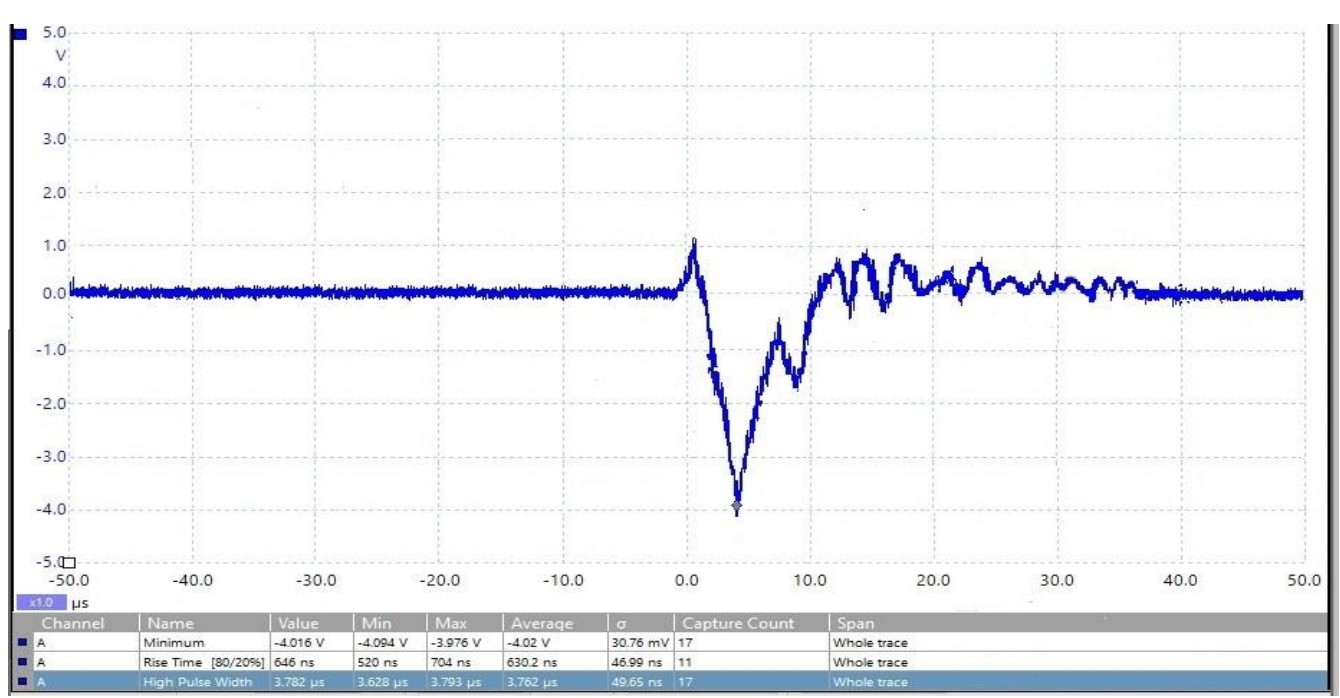

**Figure 6.** Oscilloscope snapshot of the developed Marx generator.

### 2.3. Sterilizing Washable Apparatus

The designed treatment chamber and connected plastic pipes were sterilized in a big pot of boiling water for ten minutes, ensuring that each component was dipped entirely. The equipment was dried after sterilisation before use in the experiment.

### 2.4. Preparation of the Raw Food Sample

Oranges of similar maturity levels were peeled and sliced. Fresh orange juice was extracted using a manual fruit juice extractor in a controlled, hygienic atmosphere [21]. A paper filter (MN6151/4) was used to separate the juice from the pulps and solids to avoid an electric field discharge during the PEF treatment. The filtered juice samples were put into sterilized glass bottles (250 mL), screw-capped, and refrigerated (4 °C) until further processing.

The sample's resistivity is insufficient to provide an ideal electric field if the *E. coli* population surpasses $10^7$ CFU/mL. As a result, the studies were carried out with a maximum cell population of $10^7$ CFU/mL. *E. coli* was cultured for three days on nutrient agar at 36 °C. A single *E. coli* colony was then transferred to 10 mL of tryptic soy broth and cultured for 24 h in a shaking incubator at 36 °C. Transferring 1 mL of primary *E. coli* culture into 100 mL fresh tryptic soy broth yielded a secondary *E. coli* culture. The secondary *E. coli* culture was cultivated for 24 h at 36 °C. The concentration of the *E. coli* culture was evaluated by optical density measurement at 600 nm in a spectrophotometer on the day of the experiment (Eppendorf, Germany). The sample was diluted to $10^7$ CFU/mL with a conductivity of 3.23 0.185 S/cm for electroporation tests.

### 2.5. Low-Temperature Long-Time Thermal Treated Food Sample

The most frequent thermal procedure for processing large volumes of liquid food is a high-temperature short-time pasteurisation. This technique is more efficient and quicker than low-temperature long-time (LTLT) pasteurisation. LTLT method is suitable for small-scale batch pasteurisation because the juice sample is kept in a sealed pan, which preserves the nutrients [22]. Hence, LTLT was adopted in the current research for comparison. The juice was LTLT pasteurized in a Simaco (Constant Temperature Magnetic Stirrer model 85–2 China) at $65 \pm 5$ °C for $30 \pm 1$ min [23]. This LTLT pasteurisation is suitable for small-scale batch pasteurisation because the juice sample is kept in a sealed pan. Finally, the bottles were labeled S0 for untreated samples, S1 for LTLT-pasteurized samples, and S2 for PEF-pasteurized samples. All of the tests were carried out three times in duplicate under the same processing conditions.

### 2.6. Food Sample Analysis

Chemical and microbiological examinations of treated and untreated orange juice samples were done to compare the efficacy of the proposed electroplaters with LTLT pasteurisation. These tests were carried out to measure the shelf life after 9 days of storage at 4 °C. Microbial analysis was performed in triplicate at the Institute of Bioproduct Development, UTM, using a standard technique, BP 2008, Volume IV, Appendix XVI B, on the total viable count (TVC). Microbial elimination is expressed as $\log_{10}(N/N_0)$, where $N$ represents the number of alive *E. coli* (cfu/mL) after an applied PEF condition and $N_0$ is the average number of alive *E. coli* (cfu/mL) at the beginning. Chemical analysis (color parameters, conductivity, pH, °Brix, and vitamin C) was performed in triplicate on treated and untreated juice samples at the Food and Biomaterial Engineering Research Group (FoBERG). Color measurement was done using a colorimeter Model; CR-10, Konica Minolta, Japan, while °Brix index was measured using ATAGO Pocket Refractometer, Model: Digital Hand-held PAL-1, Japan. For the pH determination, a Thermo Scientific™ Eutech™ pH 700 Meter was used, and the conductivity was measured using a DiST 4 conductivity meter, Hanna Instruments, United States. Finally, vitamin C was measured using the method described by AOAC [24]. A standard colorimetric method was established by international recommendations for photometry and colorimetry named Commission Internationale de l'Eclairage (CIE) (translated as the International Commission on Illumination). CIE L*a*b* is one standard method where the system employs three coordinates to locate a color in color space.

### 2.7. Statistical Analysis

SPSS (version 25, IBM SPSS Statistics) was used to conduct statistical analysis. Before this, the normal distribution and variance homogeneity had been examined (Shapiro–Wilk). The data were analyzed using a multivariate General Linear Model (GLM) (treatment and storage duration were treated as fixed effects), followed by Duncan's test if the ANOVA was significant ($p < 0.05$).

## 3. Results and Discussion

To figure out how much the cell temperature changes when it is pulsed, the chamber was subjected to a series of 4.5 kV, 3 μs pulses with a frequency of 5 Hz. Table 2 contains the outcomes of the study. The temperature of the sample juice rises by around 8 °C after 1500 pulses with an energy of 4500 J. This temperature rise generates an 8% impedance variation of the treatment chamber. It indicates that the voltage should be raised throughout the experiment to maintain the appropriate electric field for treating a greater number of pulses.

**Table 2.** Variation in the temperature and log reduction of the orange juice with a different number of pulses and treatment chambers.

| No. of Pulses | PEF Treatment with Sieves | | PEF Treatment without Sieves | |
|---|---|---|---|---|
| | $Log_{10}$ Reduction | Temperature Rise °C | $Log_{10}$ Reduction | Temperature Rise °C |
| 500 | 2.2 | 25 | 1.9 | 27.6 |
| 1000 | 1.8 | 29.8 | 1.8 | 32.4 |
| 1500 | 1.4 | 31.6 | 1.2 | 34.7 |

### 3.1. Microbial Analysis

Table 2 shows variations in the temperature and log reduction of the PEF-treated orange juice with different pulses and treatment chambers (with and without sieves). In both treatment chambers, the process of microbial reduction was almost linear. However, the initial 500 pulses seem to be more effective than others. PEF treatment with sieves shows a better log10 reduction (5.4). In contrast, PEF treatment without sieves shows a log10 reduction (4.9). For a total treatment of 1500 pulses, the findings of inactivation of *E. coli* show an average level of effectiveness of 37 J·mL$^{-1}$ per log decrease; without sieves, this was 48.73 J·mL$^{-1}$ per log decrease. This is a poor degree of efficiency, which might be related to the radial electric field's non-uniformity and the presence of dead zones in the treatment region.

PEF-treated juice shows improved microbial stability after treatment. The same findings were found by [25], who stated that PEF treatment is a suitable nonthermal method for the microbial inactivation of liquid foods, which produces similar inactivation to traditional thermal treatments but at room temperature. Timmermans et al. [26] found a significant inactivation of different microbial in orange juice after PEF treatment. Microbial inactivation can be increased with the higher electric field intensity and applied to the highest energy to reduce cell viability in liquid food products.

### 3.2. Chemical Analysis

Table 3 summarizes the effects of treatment and storage duration on the color, °Brix, pH, vitamin C, and conductivity of orange juice. All the samples showed a decrease in CIE L* value during storage, which indicates a darkling of juice surface color. The initial L* values were similar (~40), although this was significantly different between all treatments. However, after 9 storage days, the untreated samples presented the highest variation and, thus, the lowest L* value (36.41), while the most stable L* value was observed in PEF samples (39.40). In this case, LTLT-treated samples also suffer a significant decrease of L* value (37.94). Still, after 9 days of storage, b* values of LTLT-treated samples (26.61) and untreated samples (26.48) showed a significant decrease. In contrast, in PEF-treated samples, no significant changes were observed for this parameter (26.95).

**Table 3.** Chemical assessment of untreated, thermal (LTLT), and PEF-treated orange juice samples at Day 0 and 9 of refrigerated storage (4 °C) (mean ± SD).

| Parameters | Day 0 | | | Day 9 | | | p Values | | |
|---|---|---|---|---|---|---|---|---|---|
| | S0 | S1 | S2 | S0 | S1 | S2 | T | D | T × D |
| L* | 40.31 ± 0.06 [e] | 40.46 ± 0.03 [f] | 40.12 ± 0.04 [d] | 36.41 ± 0.04 [a] | 37.94 ± 0.03 [b] | 39.40 ± 0.04 [c] | <0.001 | <0.001 | <0.001 |
| a* | 2.04 ± 0.03 [a] | 2.13 ± 0.07 [a] | 2.18 ± 0.03 [a] | 2.63 ± 0.18 [c] | 2.54 ± 0.02 [c] | 2.36 ± 0.09 [b] | 0.375 | <0.001 | 0.006 |
| b* | 27.04 ± 0.05 [bc] | 27.17 ± 0.05 [c] | 27.07 ± 0.04 [bc] | 26.48 ± 0.29 [a] | 26.61 ± 0.60 [ab] | 26.95 ± 0.07 [abc] | 0.345 | 0.008 | 0.318 |
| °Brix | 12.40 ± 0.02 [f] | 11.93 ± 0.04 [c] | 12.34 ± 0.03 [e] | 11.07 ± 0.03 [a] | 11.67 ± 0.02 [b] | 12.27 ± 0.02 [d] | <0.001 | <0.001 | <0.001 |
| pH | 3.38 ± 0.02 | 3.36 ± 0.04 | 3.37 ± 0.03 | 3.41 ± 0.03 | 3.37 ± 0.03 | 3.38 ± 0.02 | 0.156 | 0.112 | 0.665 |
| Vitamin C [1] | 47.32 ± 0.58 [c] | 47.76 ± 0.80 [c] | 46.71 ± 0.85 [c] | 39.62 ± 0.69 [a] | 40.74 ± 0.75 [a] | 44.21 ± 0.66 [b] | 0.002 | <0.001 | <0.001 |
| Conductivity [2] | 3.17 ± 0.02 [b] | 3.13 ± 0.01 [b] | 3.16 ± 0.03 [b] | 2.97 ± 0.02 [a] | 3.13 ± 0.02 [b] | 3.13 ± 0.02 [b] | <0.001 | <0.001 | <0.001 |

S0: Untreated samples; S1: LTLT-treated samples; S2: PEF-treated samples; [1]: Vitamin C expressed as mg/100 g; [2]: Conductivity expressed as mS/cm. T: treatment effect; D: storage day effect; T × D: interaction between treatment and storage day effects; [a–f] Means without a common superscript in the same sample (for the same parameter) were considerably different ($p < 0.05$; Duncan's test).

Contrary to the behaviour of b* values, CIE a* values increased during storage in all samples. After treatment (day 0), no differences were observed between any samples, while after storage, a* value variation was lower ($p < 0.05$) in PEF samples than in LTLT or untreated samples. Thus, particularly in LTLT and untreated samples, a color shift toward negative b* and positive a* directions indicate more reddish and less yellowish color in these samples.

The total color differences (ΔE*) of untreated, thermally-treated and PEF-treated samples were 3.98, 3.59, and 0.75, respectively (data not shown). The calculated ΔE values showed a significant color change in untreated and LTHT-treated samples. However, PEF-treated samples showed a slight color change, which agrees with the earlier findings [27]. The color indicates that PEF-processed orange juices were brighter than a thermally treated sample [28]. Therefore, in our research, the application of PEF resulted in minimal changes in orange juice color. In contrast, a significant and appreciable color change occurs during storage in untreated and thermal treated samples. The color changes in orange juice produce a significant quality loss over shelf-life. Color changes may occur in response to the production of brown pigments and the fading of carotenoids, the naturally occurring pigments in orange juice [29]. The instrumental color changes could be observed in the visual appearance of orange juice. After storage, PEF-treated juice (S2) had the same physical appearance as fresh juice, while the untreated and thermally treated juices were paler. Therefore, applying PEF does not impact the juice color.

Our results agree with those reported previously by several authors [28]. A previous study reported that PEF-processed orange juice retained more color than thermally processed juice [30], which fully agrees with our results. Additionally, Cortés et al. [27] found higher color differences during refrigerated storage in the thermal pasteurized orange juice than in the juice treated by PEF.

On the other hand, the treatments and the storage affected the °Brix values. After treatments (day 0), both LTLT and PEF samples presented significantly lower values of this parameter than untreated juices, but the values were similar (between 11.93 and 12.40). These findings are consistent with those reported by other authors in the case of orange juice, which had a °Brix value of 12 [30]. Table 3 further indicates that LTLT treatment reduces the °Brix value more than PEF, which exhibited values that are quite comparable to untreated samples. The same trend was observed by Elez-Martínez et al. [31], who observed a slight decrease in the °Brix values after PEF treatment in the orange juice samples. After 9 days of storage (4 °C), a significant decrease in °Brix values was observed in all juices. This decrease was more significant in untreated samples (from 12.40 to 11.07) than in LTLT (from 11.93 to 11.67) or PEF (from 12.34 to 12.27). The higher decrease of °Brix values in untreated samples during storage agree with the results obtained by other authors and could be related to the growth of microorganisms [31], which consume soluble solids such as carbohydrates (sugars). Therefore, the results showed that PEF treatment seems to be

suitable for maintaining the °Brix values both after processing and storage. Similar findings were reported in other studies in which no differences or minimal changes in °Brix were observed in the PEF-treated orange juice during storage [30,31].

Table 3 shows that the treatment or the storage did not influence the pH values. Additionally, Elez-Martínez et al. [31] observed that the processing (PEF vs thermal) had no significant effects on pH values. Similarly, in another study, the authors did not find pH differences between untreated, thermal, or PEF-treated orange juice [25]. In contrast, Cortés et al. [27] found that the pasteurisation treatment significantly decreased pH values, although the difference was very low. Additionally, the same authors also reported an increase in pH value during juice storage and attributed this fact to the microbial deterioration of juice.

Vitamin C content did not change after processing (day 0) in any of the treatments used in the present study (about 47 mg/100 g). This value was similar to what Agcam et al. [32] reported. In contrast to our findings, some other authors reported that the thermal treatment significantly impacted the vitamin C content [31]. However, it is important to highlight that the thermal treatments of the other studies used higher temperatures (90–95 °C) and lower times (15–90 s) than those employed in the present research (65 °C for 30 min), which could explain the differences observed. As expected, the vitamin C content decreased during storage. This decrease was more significant in untreated and LTLT samples (from ~47 to ~40 mg/100 g) than in the PEF-treated samples (46.71 to 44.21 mg/100 g). Thus, vitamin C content in thermally treated samples was reduced by 14% after storage of 9 days, and in the PEF-treated sample it was reduced to 6.6% after storage. Several studies demonstrated that PEF treatment is suitable for maintaining the vitamin C content during storage [33–35], which agrees with the present findings.

Finally, the influence of PEF on the conductivity of orange juices than thermally treated and fresh juices was determined. Both treatments showed no discernible impact on the conductivity of the juice samples after treatment (day 0). However, after nine days of storage, the conductivity of untreated samples dramatically reduced, while the treated samples (LTLT and PEF) did not change. This may be related to the excessive development of bacteria in the sample, which caused a significant alteration in the liquid's chemical structure, thus reducing ion mobility and resulting in a lower conductivity value [33–35].

### 3.3. Energy Consumption Estimation

Energy losses occur during the conversion and depend on the process and individual equipment design. The total energy was 20 kJ·sec$^{-1}$ for PEF treatment from the above calculation. The heating power of the 85–2 hot plate magnetic stirrer was 250 W (250 J·sec$^{-1}$), since the heating plate was used for 30 min. Hence, the total energy consumed by LTLT is almost double that of the proposed PEF treatment.

### 4. Conclusions

This research covers the essential process parameters and the technique for designing the essential elements of a PEF pasteurisation system. Some measures were taken to maintain the electrical resistance of the coaxial treatment chamber. The electrical resistance of the treatment chamber controls the output of the pulse generator. The newly designed coaxial treatment chamber developed an IGBT-based Marx generator to pasteurize orange juice. The proposed electroporator system was tested for pasteurisation efficiency using orange juice, and the results were compared to those pasteurized with LTLT. PEF treatment with static mixers makes the microbial spoilage effectively inactive, similar to thermal treatment. Additionally, PEF processing did not cause degradation of the quality attributes, particularly the color, appearance, vitamin C content, or °Brix value of orange juice. In conclusion, the PEF processing extended the shelf life of orange juice while not negatively affecting the quality parameters, in contrary to thermal processing. Furthermore, PEF treatment consumes almost half of the electric energy compared to LTLT thermal treatment. Therefore, PEF technology is a sustainable solution to preserve the desirable characteristics

of orange juice and extend its shelf life. Future research, however, is required to design treatment chambers with higher electrical resistance to design power supplies. The findings reported in this study will aid future research into PEF to enhance the electroporator. This study provides significant findings in developing and utilizing PEF-based food treatment to inactivate microorganisms, particularly at low liquid flow speed. Hence, it may be further developed for industrial applications. The development of the proposed electroporator is suitable for promoting the successful transfer of scientific knowledge on PEF effects on fruit juices to the food industry.

**Author Contributions:** Conceptualisation, R.N.A., Z.A.-M. and R.M.A.; formal analysis, R.N.A., Z.A.-M. and Y.M.M.J.; data curation, R.N.A., Z.A.-M., P.T., E.R., R.M., J.M.L. and R.M.A.; writing—original draft preparation, R.N.A., Z.A.-M., R.M.A., J.M.L. and E.R.; writing—review and editing, P.T., R.M. and R.M.A. All authors have read and agreed to the published version of the manuscript.

**Funding:** This research was funded by Universiti Teknologi Malaysia through votes 04G81, 07G05, 16J61, 01M73, 4B482, 05G88, and 4B383.

**Data Availability Statement:** All data are presented in the article.

**Acknowledgments:** Rai Naveed Arshad is thankful to Universiti Teknologi, Malaysia for conducting his research.

**Conflicts of Interest:** The authors declare no conflict of interest.

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
