# Peer review of "Sustainable Electroporator for Continuous Pasteurisation: Design and Performance Evaluation with Orange Juice"

_sustainability, doi:10.3390/su14031896_

Round 1
Reviewer 1 Report
Thank the authors for sending back the revised manuscript and it was very different from what I saw last time.
And I believe the authors provide with explanations of avoiding self-plagiarism and corrected all the issues I pointed out last time.
Author Response
Reviewer 1
Comments and Suggestions for Authors
Thank the authors for sending back the revised manuscript and it was very different from what I saw last time.
And I believe the authors provide with explanations of avoiding self-plagiarism and corrected all the issues I pointed out last time.
Answer: Thanks a lot for your time to revise this paper. Dear Prof; overall similarity index is well under acceptable range and single source as well.
Reviewer 2 Report
Dear authors,
This manuscript reports on the results of developing a simple, affordable, and portable electroporator for liquid food pasteurization. The study presents further electroporators for other food applications with different treatment 37 chambers without compromising the product's quality. The authors are advised to highlight what makes your manuscript different than those already published?
It is clear that this article has been well drafted, and the authors made some modifications, and I think that the research in this form and content is good for publication in the journal.
Additional comments are provided below to help
improve the manuscript.
- Lin 22-24: I recommend being “Despite its wide range of applications, electroporators are out of reach for many labs due to their high development costs, and different electroporators have been tailored to specific applications.
- Line 254: E. coli should be italic “E. coli” and please correct all over the manuscript.
- Line 251; Food sample analysis section: The authors should write how they conduct the microbial examination, so that the experiment can be repeated in the future.
Regards,
Author Response
Reviewer 2
Comments and Suggestions for Authors
Dear authors,
This manuscript reports on the results of developing a simple, affordable, and portable electroporator for liquid food pasteurization. The study presents further electroporators for other food applications with different treatment chambers without compromising the product's quality. The authors are advised to highlight what makes your manuscript different than those already published?
Answer: Thanks a lor for your comments, we have revised the paper according to your comments.
It is clear that this article has been well drafted, and the authors made some modifications, and I think that the research in this form and content is good for publication in the journal.
Answer:
Additional comments are provided below to help improve the manuscript.
Line 22-24: I recommend being “Despite its wide range of applications, electroporators are out of reach for many labs due to their high development costs, and different electroporators have been tailored to specific applications.
Answer: Thanks. Corrected as suggested.
Line 254: E. coli should be italic “E. coli” and please correct all over the manuscript.
Answer: Modified throughout the manuscript.
Line 251; Food sample analysis section: The authors should write how they conduct the microbial examination, so that the experiment can be repeated in the future.
Answer: Thanks prof for your concern. It has been included in 2.3 section.